# “No-Reflow” Phenomenon: A Contemporary Review

**DOI:** 10.3390/jcm11082233

**Published:** 2022-04-16

**Authors:** Gianmarco Annibali, Innocenzo Scrocca, Tiziana Claudia Aranzulla, Emanuele Meliga, Francesco Maiellaro, Giuseppe Musumeci

**Affiliations:** Cardiology Department, Azienda Ospedaliera Ordine Mauriziano Umberto I, 10128 Turin, Italy; gannibali@mauriziano.it (G.A.); iscrocca@mauriziano.it (I.S.); taranzulla@mauriziano.it (T.C.A.); emeliga@mauriziano.it (E.M.); francescomll91@gmail.com (F.M.)

**Keywords:** myocardial infarction, no-reflow, percutaneous coronary intervention, acute coronary syndrome

## Abstract

Primary percutaneous angioplasty (pPCI), represents the reperfusion strategy of choice for patients with STEMI according to current international guidelines of the European Society of Cardiology. Coronary no-reflow is characterized by angiographic evidence of slow or no anterograde epicardial flow, resulting in inadequate myocardial perfusion in the absence of evidence of mechanical vessel obstruction. No reflow (NR) is related to a functional and structural alteration of the coronary microcirculation and we can list four main pathophysiological mechanisms: distal atherothrombotic embolization, ischemic damage, reperfusion injury, and individual susceptibility to microvascular damage. This review will provide a contemporary overview of the pathogenesis, diagnosis, and treatment of NR.

## 1. Introduction

Cardiovascular diseases and, in particular, acute myocardial infarction with ST-segment elevation (STEMI) represent a major cause of mortality in industrialized countries. Primary percutaneous angioplasty (pPCI) represents the reperfusion strategy of choice for patients with STEMI according to current international guidelines of the European Society of Cardiology (ESC) [1]. However, even after the restoration of culprit vessel patency, suboptimal coronary reperfusion, less than three according to the Thrombolysis in Myocardial Infarction (TIMI) score, may occur, with slow, incomplete, or absent coronary flow in the affected coronary artery [2]. This phenomenon, which can regress spontaneously in about half of the cases, is called “no-reflow” (NR) or microvascular obstruction (MVO), and can complicate up to 60% of STEMI cases [1,3]. NR can occur in both the setting of acute coronary syndrome and in the stable patient and is due to a structural and functional alteration of the coronary microcirculation. In addition, it is associated with an increased incidence of rehospitalization, negative ventricular remodeling, malignant arrhythmias, and heart failure and is an independent predictor of myocardial infarction and death [4,5,6]. Among the risk factors, we can list cardiovascular risk factors such as: an age over 65 years, hypertension, smoking, dyslipidemia, diabetes, renal failure, inflammatory processes, and a history of atrial fibrillation, and procedure-related factors such as: the presence of an increased thrombotic load, delayed presentation, high-pressure inflations, and the use of debulking devices [7,8].

## 2. Pathophysiological Mechanisms

NR is related to a functional and structural alteration of the coronary microcirculation and we can list four main pathophysiological mechanisms: distal atherothrombotic embolization, ischemic damage, reperfusion injury, and individual susceptibility to microvascular damage [9] (Figure 1). A complex atherosclerotic plaque can lead to distal embolization phenomena both during the acute and procedural phases, leading to increased distal vascular resistance and additional microinfarcts that promote the release of pro-inflammatory and vasoconstrictive substances [10,11]. The severity of ischemic injury is directly proportional to the duration of ischemia time. Ischemic damage results in the death of cardiomyocytes, endothelial cells, and formation of interstitial edema with impaired nitric oxide production and subsequent microcirculation obstruction favored by vascular endothelial growth factors (VEGF) release that increase vascular permeability [12,13]. Reperfusion injury, on the other hand, is caused by the abrupt restoration of blood flow at the level of the damaged microcirculation, causes direct cardiomyocyte damage with an influx of inflammatory neutrophils during reperfusion that promotes the production of inflammatory cytokines, free oxygen radicals, vasoactive substances, and proteolytic enzymes [14,15]. The presence of preexisting endothelial dysfunction or genetic mutations, such as the 1976TC polymorphism of the gene for adenosine receptors and various ion channels, increases the susceptibility to microvascular dysfunction and no-reflow [16,17].

## 3. Diagnosis of No-Reflow

Coronary angiography during pPCI is the most frequently used diagnostic method for the diagnosis of NR that, thanks to the use of TIMI flow classification, allows to classify coronary flow on a scale from 0, absence of flow, to 3, presence of normal flow [2]. Next to this, we also have the TIMI frame count that evaluates the number of frames required for the contrast agent to fill the distality of the coronary arteries. An increased number of frames constitutes an indirect index of NR [18]. However, because of the poor sensitivity and specificity of this, another angiographic assessment was subsequently introduced by evaluating the degree of myocardial “blush” (MBG). Blush, in fact, assesses the intensity of myocardial tissue radiopacity obtained by injection of contrast medium into the epicardial coronary arteries and the rapidity with which this impregnation decreases [19] (Table 1). MBG also ranges from 0 to 3 and is diagnostic of NR for values of 0–1 [1].

Instead, a more accurate invasive assessment is possible through flow parameters or resistance parameters. Coronary flow reserve (CFR), in fact, through the ratio of coronary flow during maximal hyperemia to coronary flow at rest, provides information about the microcirculation in the absence of epicardial stenosis. A value < 2.0 was associated with the presence of MVO with a sensitivity of 79%. In addition, the measurement of coronary blood flow velocity using intracoronary Doppler guidance allows for the detection of the typical flow pattern associated with NR, characterized by early retrograde systolic flow and rapid deceleration of diastolic flow [20,21]. The microvascular resistance index (IMR), based on the principle of thermodilution, is defined as the product of distal coronary pressure and the mean transit time of a bolus during maximum hyperemia using a dual pressure and temperature guide, and provides an assessment of microcirculation independent of hemodynamic parameters. IMR values > 25 correlate with the presence of MVO, and a post-procedure IMR > 40 units has been associated with a higher rate of in-hospital adverse events, mortality, and readmission for heart failure at 1-year follow-ups [22,23]. Alternatively, IMR under conditions of maximal hyperemia incorporates Doppler flow velocity to estimate flow, and values > 2.5 mmHg/cm/s are predictive of MVO [21,24]. Recently, angiography-derived IMR (IMR_angio_) has been studied, with good diagnostic accuracy in predicting both IMR > 40 units and the presence of large MVOs on cardiac MRI being documented [25]. Another new technology is CorFlow Therapy™ (CoFl™), which combines real-time microvascular assessment with the ability to administer intracoronary drugs [26]. This device determines transient coronary occlusion by balloon inflation, incremental infusions of crystalloid at a predefined flow rate, and simultaneous measurement of distal pressure beyond balloon occlusion. The flow and pressure quotient can be used to derive dynamic microvascular resistance and have real-time diagnosis of microvascular dysfunction. Initially validated in a porcine model, early results from the MOCA I trial are encouraging in terms of safety, applicability, and the ability to detect MVO immediately after pPCI [27].

Gadolinium-enhanced cardiovascular magnetic resonance imaging (MRI) is certainly the “gold standard” for NR diagnosis [1]. A 1% increase in the extent of MVO is associated with a 1.14-fold increased risk of 1-year mortality [6]. Coronary microvasculature becomes occluded due to the presence of erythrocytes, neutrophils, and cellular debris resulting in a lack of gadolinium enhancement in the endocardial nucleus [28]. The cardiac magnetic resonance (CMR) allows the visualization of myocardial damage through the use of different techniques including delayed gadolinium contrast enhancement (DGE) and T2-weighted images [29]. In addition, new parametric mapping techniques allow for the accurate quantification of myocardial damage based on changes in T1, T2, T2* release times and the assessment of extracellular volume [29]. T2 sequences, in addition to being critical for discriminating between acute and chronic myocardial infarction (generally, edema dissolves in approximately 4–6 weeks after infarction), allow for the identification of areas of intramyocardial hemorrhage (IMH) [30,31]. IMH is a strong predictor of left ventricular remodeling independent of infarct area, and it is closely associated with adverse outcomes. On T2-weighted images, areas of IMH appear of attenuated signal within high-signal edematous areas because of the presence of hemoglobin degradation products. The identification of areas of MVO requires the use of the contrastographic technique [32].

Gadolinium has an extravascular and extracellular distribution, so its wash-out is delayed in areas of increased extracellular/interstitial volume, such as areas of necrosis (in the acute phase) and fibrosis (in the chronic phase) [33]. DGE is assessed in T1-weighted images 10–15 min after gadolinium administration and is used to visualize the MVO, which appears as a dark, hypointense area surrounded by the hyperintensity of necrotic myocardium [32]. Alternatively, early contrastographic impregnation is a contrast-dependent technique in which T1-weighted acquisitions are performed just after contrast medium administration (after 1–3 min). Low-signal areas represent areas of MVO or thrombus. Finally, the first-pass perfusion (FPP) method is another contrast-dependent technique that allows the detection of even small areas of MVO [34]. FPP is a dynamic study and is based on the visualization of the time distribution of the bolus of the paramagnetic contrast agent during the first pass at the level of the myocardial microcirculation [33]. A perfusion defect is thus manifested as a region of contrastographic failure to impregnate myocardial tissue due to altered capillary microcirculation. However, the prognostic value of FPP is not as strong as for DGE, presumably in view of the fact that it also detects small areas of MVO [34].

ECG may also allow for a diagnosis of NR to be made. A resolution of ST-segment elevation <50% or <70%, depending on the cut-off used, after 60 to 90 min after reperfusion is indicative of NR [35,36].

Other diagnostic techniques used to assess NR are contrast-enhanced echocardiography and nuclear imaging with positron emission tomography and single-photon emission computed tomography [37,38,39]. Contrast-enhanced echocardiography is an examination that can be performed at the patient’s bedside in which microbubbles of inert gas are typically administered intravenously, and NR is identified by areas of hypoperfusion [39,40]. However, the lack of sensitivity and/or the complexity of implementation make these techniques less attractive for the evaluation routine assessment of NR [1,41].

**Table 1 jcm-11-02233-t001:** Summarizes the main diagnostic methods available and their limitations.

Diagnostic Methods	Study Design	Results	Limitations
Coronary Angiography (MBG) [42]	777 prospectively enrolled patients who underwent pPCI during a 6-year period.	MBG can be used to describe the effectiveness of myocardial reperfusion and is an independent predictor of long-term mortality.	Interobserver and intraobserver variabilities associated with subjective angiographic assessments.
Coronary Flow Reserve (CFR) [43]	89 prospectively enrolled patients who underwent pPCI during a 4-year period and subsequent physiologic study.	A CFR value ≥ 2.0 is considered normal. Complimentary assessment of microcirculation by the IMR and CFR may be useful to evaluate myocardial viability and predict the long-term prognosis of STEMI patients.	Possible significant variability of tracings between different beats. Does not distinguish between epicardial and microvascular components of coronary resistances. Requires maximal hyperemia using adenosine.
Microvascular resistance index (IMR) [44]	288 prospectively enrolled patients with STEMI during a 11-year period.	An IMR > 40 is a multivariable associate of left ventricular and clinical outcomes after STEMI, regardless of infarct size. IMR has superior clinical value for risk stratification.	Manual injection of saline may be a source of variability. It requires achievement of maximal hyperemia and the use of adenosine.
Electrocardiogram (ECG) [36]	180 prospectively enrolled patients with a first acute STEMI.	Residual ST-segment elevation and the number of Q waves on the ECG shortly after pPCI have complementary predictive value on myocardial function, infarct size and extent, and MVO.	Discordance between resolution of ST-segment elevation and the angiographic indices of NR.
Myocardial Contrast Echocardiography (MCE) [40]	110 prospectively enrolled patients who underwent pPCI in a multicenter study.	Among patients with TIMI 3 flow, MVO extension, as detected and quantified by MCE, is the most powerful independent predictor of LV remodeling after STEMI compared with persistent ST-segment elevation and degree of MBG.	Operator-dependent and limited by the possible poor acoustic window.
Cardiac Magnetic Resonance (CMR) [6]	Pooled analysis using individual patient data from seven randomized primary PCI trials	The presence and extent of MVO measured by CMR after primary PCI in STEMI are strongly associated with mortality and hospitalization for HF within 1 year.	Usually performed 2 to 7 days after pPCI. Not widely available locally. Not performable in all patients.
Positron Emission Tomography (PET) [37]	Seven porcine model with left anterior descending coronary artery occlusion/reperfusion underwent PET-CT within 3 days of infarction.	Increased regional FDG uptake in the area of acute infarction is a frequent occurrence and indicates tissue inflammation that is commonly associated with MVO.	Expensive and difficult to obtain locally.

pPCI, Primary Percutaneous Coronary Intervention; MBG, Myocardial Blush Grade; STEMI, ST-Elevation Myocardial Infarction; NR, No-Reflow; CMR, Cardiac Magnetic Resonance; MCE, Myocardial Contrast Echocardiography; TIMI, Thrombolysis in Myocardial Infarction; MVO, Microvascular Obstruction; LV, Left Ventricular; HF, Heart Failure; PET, Positron Emission Tomography; PET-CT, Positron Emission Tomography/Computed Tomography; and FDG, 2-Deoxy-2-[18F]Fluoro-d-Glucose.

## 4. Management of No-Reflow

Although NR has been a known phenomenon for many years, the efficacy of therapies in animal models has only partially translated to humans with benefits on surrogate endpoints but no impact on endpoints such as cardiovascular mortality. To date, the main treatment of NR is based on the use of intracoronary drugs that can result in vasodilation in the coronary arteries. Several studies have shown possible efficacy for vasodilator drugs, such as adenosine, calcium channel blockers, and sodium nitroprusside, used singularly or in combination, and antiplatelet drugs such as glycoprotein IIB/IIIA inhibitors. Alongside these, nonpharmacologic treatment strategies such as coronary post-conditioning, remote ischemic conditioning, or tools to reduce the embolization of thrombotic material and increase coronary flow have also been investigated in several trials, but there is still no therapy, single or in combination, aimed at reducing ischemia/reperfusion injury that is clearly associated with improved clinical outcomes [1,45].

In the following paragraphs, we will review the main treatment strategies currently available and future ones under evaluation in different trials.

## 5. Pharmacological Treatment

### 5.1. Β-Blockers

The effect of this class of drugs has been primarily studied in terms of cardiomyocyte protection and infarct extension. In some animal models, however, metoprolol, before reperfusion, reduced the size of the infarct area and the occurrence of NR with an anti-inflammatory action through inhibition of neutrophil-platelet aggregate formation [46]. In the METOCARD-CNIC (Effect of Metoprolol in Cardioprotection During an Acute Myocardial Infarction) study, metoprolol, administered before pPCI and through a time-dependent action, reduced the extent of infarction, prevented adverse left ventricular remodeling, preserved systolic function, and reduced the rate of rehospitalization for heart failure [47]. A sub analysis of this study also documented an interaction between metoprolol and neutrophil count with a modulating effect of metoprolol on neutrophil impact on MVO [48].

Less encouraging data, however, are from the EARLY-BAMI (Early-Beta Blocker Administration Before Reperfusion Primary PCI in Patients With ST-Elevation Myocardial Infarction) trial, which failed to document a reduction in infarct extension at 1 month in patients treated with intravenous metoprolol before pPCI [49]. Reasons given included different drug dosage, timing of administration, and the patient population under investigation [48].

In contrast, other molecules such as carvedilol and nebivolol have demonstrated protection of the coronary microcirculation in preclinical studies [19].

Current guidelines from ESC recommend the use of intravenous beta-blockers in STEMI patients undergoing pPCI without signs of acute heart failure and with systolic blood pressure > 120 mmHg (recommendation class IIa, level of evidence A) [1].

### 5.2. Calcium Channel Blockers

Calcium channel blockers (CCBs) (verapamil, diltiazem, nicardipine) are used to treat no-reflow through various mechanisms. Through channel binding on vascular smooth muscle, cardiac myocytes, and nodal cells, they result in smooth muscle relaxation and coronary vasodilation. Several studies, with numerous limitations of selection and measurements, have demonstrated benefits in NR treatment for verapamil and diltiazem with better outcomes in those treated intracoronary [50,51,52]. In particular, nicardipine has documented better outcomes in combination with rotational atherectomy for the prevention of no-reflow [53]. However, to date, data on CCBs are insufficient to show significant beneficial effects on no-reflow.

### 5.3. Adenosine

Adenosine is a purine nucleoside with a short half-life (<2 s) and numerous pleiotropic effects including vasodilation of the coronary microcirculation via binding to A2 receptors and smooth muscle relaxation. It also has anti-inflammatory properties against neutrophils and inhibition of platelet aggregation, promotes ischemic preconditioning by limiting reperfusion injury, and exhibits anti-apoptotic and pro-angiogenic effects. Side effects include bradycardia with atrioventricular block, hypotension, dyspnea, bronchospasm, and flushing [54].

The REOPEN-AMI (Intracoronary Nitroprusside Versus Adenosine in Acute Myocardial Infarction) trial documented a significant improvement in MVO and peak troponin compared with placebo or sodium nitroprusside, leading to a reduction in major cardiovascular events and favorable left ventricular remodeling at 1 year after the event [55].

Data in contrast to early studies on the use of adenosine after pPCI, (AMISTAD [56] and AMISTAD-II [57]) which documented a reduction in infarct size in adenosine-treated patients without significant differences in clinical outcomes and from the REFLO-STEMI (Reperfusion Facilitated by Local Adjunctive Therapy in ST-Elevation Myocardial Infarction) study, also showed potential harmful effects [58].

### 5.4. Sodium Nitroprusside

Sodium nitroprusside is a non-selective drug metabolized to its active form, nitric oxide, that acts as a potent vasodilator in the coronary and peripheral microcirculation and by inhibiting platelet aggregation. Its latency of action appears to be more prolonged than other vasodilators [59,60]. Furthermore, in comparison with drugs such as tirofiban, it has demonstrated a lower rate of adverse events, an improvement in TIMI frame count, a more rapid resolution of ST-segment elevation, and a higher rate of left ventricular ejection fraction without achieving a significant difference in TIMI grade [61]. Although there are no data to support the preventive capacity of NR, nitroprusside, at 6-month follow-up, documented lower rates of revascularization, myocardial infarction, or death compared with placebo-treated patients. However, further studies are needed for a more accurate assessment of the ability of nitroprusside to prevent NR [62].

### 5.5. Epinephrine

Among the pharmacological alternatives available is also intracoronary epinephrine, a drug with limited experience compared to others [63,64] but which has recently shown encouraging results for the treatment of NR refractory to other therapies or where these could not be used. In 2020, the RESTORE trial, a multicenter observational study, was published to evaluate the safety and efficacy of epinephrine in NR during STEMI compared with conventional therapy. Navarese et al. documented a significant improvement in coronary flow, left ventricular ejection fraction, ST-segment resolution, and clinical events at 30 days in STEMI patients with refractory NR compared with the control group [65]. More recently, the COAR trial, an open-labeled study that randomized patients to intracoronary epinephrine vs. adenosine, was published, demonstrating improved end coronary flow and relative safety of epinephrine in normotensive patients with acute coronary syndrome [66]. The main side effect of epinephrine is the risk of developing malignant arrhythmias [67]. Figure 2 represents a case of no refractory reflow management handled at our center.

### 5.6. Nicorandil

Nicorandil is a vasodilator drug that acts through potassium channels and intracellular cGMP concentrations. It is used for the treatment of angina pectoris during acute coronary syndromes in Japan and some other Asian and European countries because it showed improved coronary perfusion and lower no-reflow rates in a previous meta-analysis [68].

### 5.7. Antiplatelet Therapy

With regard to the major antiplatelet drugs, no NR or myocardial perfusion benefits were documented from sub-analysis of the PLATO (Study of Platelet Inhibition and Patient Outcomes) in the ATLANTIC (Administration of Ticagrelor in the Cath Lab or in the Ambulance for New ST Elevation Myocardial Infarction to Open the Coronary Artery) study and in the REDUCE-MVI (Reducing Micro Vascular Dysfunction in Acute Myocardial Infarction by Ticagrelor) study [69,70,71]. The PLEIO study, however, recently showed superior recovery of microcirculation function in patients treated with ticagrelor compared with clopidogrel [72]. This is in line with a previous meta-analysis that demonstrated a greater benefit of ticagrelor over clopidogrel in reducing NR and incidence of MACE without significantly increasing the risk of bleeding [73].

Among novel antiplatelet agents, we are awaiting data from the Platelet Inhibition to Target Reperfusion Injury (PITRI) trial, which evaluated the ability of cangrelor, administered before reperfusion, to reduce the size of acute myocardial infarction and MVO by CMR [74].

Glycoprotein IIB/IIIA inhibitors are potent antiplatelet agents that inhibit platelet aggregation and have demonstrated benefit in the era before the routine use of dual antiplatelet therapy [1]. To date, there have been no studies showing convincing benefits of glycoprotein IIB/IIIA inhibitors in addition to standard therapy [75]. However, the On-TIME-2 (Ongoing Tirofiban in Myocardial Infarction Evaluation 2) study showed that prehospital initiation of bolus tirofiban could result in ST-segment resolution and improve clinical outcome after pPCI [42]. Additional conflicting data are those regarding the route of administration, intracoronary or intravenous. Although the CICERO trial (Comparison of Intracoronary Versus Intravenous Abciximab Administration During Emergency Reperfusion of ST-Segment Elevation Myocardial Infarction) and INFUSE-AMI study (Intracoronary Abciximab and Aspiration Thrombectomy in Patients with Large Anterior Myocardial Infarction) reported benefits in terms of reduction of infarct area after intracoronary administration of abciximab, the AIDA STEMI study (Abciximab Intracoronary versus intravenous Drug Application in STEMI) documented similar rates of major adverse cardiovascular events after 90 days and 1 year between the two modalities of administration [76,77,78]. Recently, in a series of 71 STEMI cases treated with pPCI, the combined use of glycoprotein IIB/IIIA inhibitors along with aspiration and balloon inflation resulted in decreased NR rates [79].

Finally, according to current ESC guidelines, GP IIb/IIIa inhibitors should be considered (class of recommendation IIa, level of evidence C) if there is evidence of NR or thrombotic complication [1].

### 5.8. Intracoronary Fibrinolysis

The role of fibrinolytic therapy is also still under study. In fact, although some initial encouraging data documented benefits, in terms of myocardial reperfusion, subsequent studies have not confirmed these data [80]. Among them, the randomized T-TIME trial recently demonstrated that low-dose intracoronary alteplase does not improve MVO [81]. Therefore, at present, current data do not support its use as adjuvant therapy to improve NR [48]. Recently, however, a meta-analysis by Alyamani et al. showed that a targeted thrombolytic IC approach seems safe and able to increase the efficacy of pPCI [82].

### 5.9. Statins

Statin therapy, probably through pleiotropic effects independent of the effect on lipid metabolism, also seems to have beneficial effects in the treatment and prevention of NR [9]. In the STATIN STEMI (Efficacy of High-Dose AtorvaSTATIN Loading Before Primary Percutaneous Coronary Intervention in ST-Elevation Myocardial Infarction) study, high doses of statins improved angiographic MVO but not infarct extension, compared with low doses [83]. Data was also confirmed by the SECURE-PCI (Statins Evaluation in Coronary Procedures and Revascularization) study, which showed an almost 50% reduction in cardiovascular events at 30 days with high-dose atorvastatin compared with placebo [84]. In addition, statin therapy already on board at the time of the event reduced NR rates, improved myocardial functional recovery at follow-up, and reduced the extent of infarction compared with naive patients.

All the main pharmacological treatments of NR and their side effects are resumed in Table 2.

Figure 3 shows an algorithm of management and treatment of the no-reflow phenomenon applied at our center.

## 6. Non-Pharmacological Treatment

### 6.1. Ischemic Conditioning

Ischemic preconditioning is the most powerful endogenous mechanism capable of reducing the extent of myocardial infarction by cycles of coronary balloon occlusion and reperfusion [5]. However, although the recent CONDI-2/ERIC-PPCI study did not demonstrate the efficacy of ischemic preconditioning on clinical endpoints [85], as already shown in other large trials [26], a recent randomized trial showed encouraging results regarding the incidence of NR by prolonged balloon inflation during stent deployment [86]. Although ischemic postconditioning has been shown to reduce no-reflow in small studies [87], larger randomized trials of POST, DANAMI-3-iPOST, POSTEMI, and LIPSIA CONDITIONING have not supported its choice in clinical practice [26].

### 6.2. Thrombus Aspiration

Thrombus aspiration or coronary filters are tools designed to reduce distal embolization injury, which is one of the etiopathogenetic mechanisms of NR [41].

However, the routine use of thrombus aspiration, initially associated with better clinical outcomes in STEMI patients [48], has been progressively downgraded because of its inability to reduce 30-day mortality in trials in subsequent years [88,89]; it is even contraindicated as a routine maneuver in the most recent ESC guidelines (recommendation class III) [1].

Another mechanical approach to reduce distal embolization during STEMI consists of the placement of filters, devices placed before stent deployment, which, however, have never documented an effective improvement in microvascular flow, infarct extension, or clinical outcomes [41].

Among the devices tested, we also find the pressure-controlled intermittent coronary sinus occlusion (PICSO). This is a device of transient occlusion of the flow in the coronary sinus with the aim of increasing cardiac venous pressure and thus improve the perfusion of the microcirculation [90]. The OxAMI-PICSO (Oxford Acute Myocardial Infarction-Pressure-Controlled Intermediate Coronary Sinus Occlusion) trial tested the use of PICSO prior to stent release in patients with IMR > 40, demonstrating less extension of infarction at 6 months in patients treated with PICSO compared with the control group [91].

### 6.3. Future Perspectives

Among the techniques being tested in recent years, we can list therapeutic hypothermia, which has shown favorable results in animals but controversial results in humans [92]. Although CHILL-MI, the VELOCITY trial, and COOL-AMI EU showed no benefit but rather an increase in adverse events without a reduction in infarct size or MVO, we are awaiting data from the randomized EURO-ICE trial to further evaluate the efficacy of alternative cooling technologies in NR [93,94,95,96].

Another technique is hyperoxemic reperfusion, recently approved by the FDA, which consists of the administration of supersaturated oxygen for 90 min after completion of PCI in patients with STEMI. This technique, evaluated in the AMIHOT I and AMIHOT II studies, documented a reduction in final infarct size in spite, however, of an increase in bleeding [97,98]. Therefore, in a more recent trial, the IC-HOT study, an infusion of hyperoxygenated blood at the origin of the left main with a duration of about 60 min was used [99].

New therapeutic targets could also be those represented by the modulation of the inflammatory response. Possibly, a “tailored” anti-inflammatory approach in patients with evidence of myocardial edema at CMR could benefit this subgroup of individuals [9].

Finally, other possible “cellular” approaches could be represented by pericytes, which are responsible for vasomotility in the coronary microcirculation [100], and by stem cells, exploiting the photobiostimulatory effects of low-level laser therapy that promotes the recruitment of mesenchymal stem cells into the myocardium [101].

## 7. Conclusions

NR is an event that complicates approximately 30–60% of pPCI and is associated with adverse clinical events in patients with STEMI. It is a complex phenomenon, dependent on several etiopathogenetic mechanisms, often combined, that puts a strain on the interventional cardiologist. Although it is a known phenomenon and has been studied for several years, there is currently no treatment that has demonstrated clear efficacy in terms of reduction of clinical adverse events. Given its multifactorial nature, a combined approach, using both pharmacologic and nonpharmacologic treatments, could be the strategy to pursue to improve the prognosis of these patients.

## Figures and Tables

**Figure 1 jcm-11-02233-f001:**
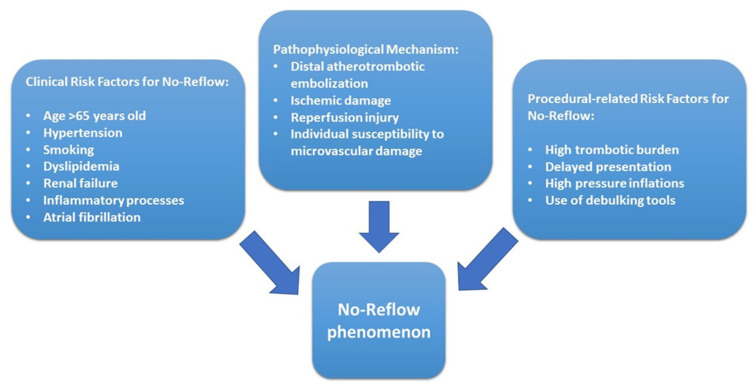
Summary of all the different factors involved in the genesis of the NR phenomenon.

**Figure 2 jcm-11-02233-f002:**
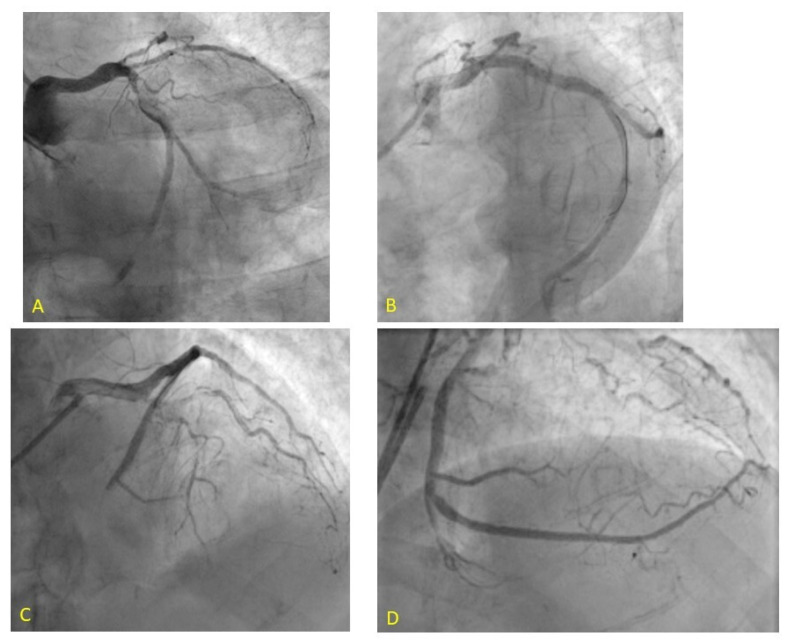
A case of refractory no-reflow management managed at our center. A 93-year-old patient with inferior-posterior STEMI and thrombotic sub occlusion of left circumflex coronary artery (**A**). After successful thrombus aspiration and stenting (**B**), no-reflow phenomenon refractory to the use of adenosine and sodium nitroprusside with development of bradycardia and hypotension (**C**) resolved after administration of intracoronary epinephrine (**D**).

**Figure 3 jcm-11-02233-f003:**
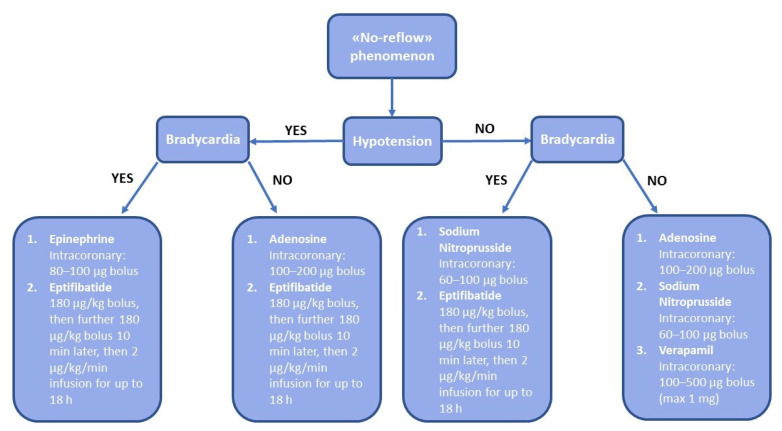
Algorithm of management and treatment of the no-reflow phenomenon applied at our center.

**Table 2 jcm-11-02233-t002:** Main drugs and dosages for the treatment of No-Reflow.

Medication	Dosage	Side Effects
Adenosine	Intravenous: 70 μg/kg/min infusion Intracoronary: 100–200 μg bolus	Bradycardia, hypotension, chest pain, dyspnea
Sodium Nitroprusside	Intracoronary: 60–100 μg bolus	Bradycardia and hypotension
Verapamil	Intracoronary: 100–500 μg bolus (max 1 mg)	Bradycardia, transient heart block
Diltiazem	Intracoronary: 400 μg bolus (max 5 mg)	Bradycardia, hypotension
Nicardipine	Intracoronary: 200 μg (max 1 mg)	Bradycardia, hypotension
Epinephrine	Intracoronary: 80–100 μg bolus	Malignant arrhythmias
Nicorandil	500 μg (max: 5 mg)	Malignant arrhythmias
Streptokinase	250 kU over 3 min	Bleeding
Tenecteplase	5 mg (max: 25 mg)	Bleeding
Tissue plasminogen activator (tPA)	0.025–0.5 mg/kg/h	Bleeding
Abciximab	0.25 mg/kg bolus, then 0.125 μg/kg/min (max 10 μg/min) infusion for 12 h	Bleeding
Eptifibatide	180 μg/kg bolus, then further 180 μg/kg bolus 10 min later, then 2 μg/kg/min infusion for up to 18 h. If CrCl < 50 mL/min, reduce infusion by 50%	Bleeding
Tirofiban	25 μg/kg over 3 min, then 0.15 μg/kg/min infusion for up to 18 h If CrCl < 30 mL/min, reduce infusion by 50%	Bleeding

CrCl: creatinine clearance.

## Data Availability

The study does not report any data.

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
