# Peer review of "“No-Reflow” Phenomenon: A Contemporary Review"

_jcm, 2022, doi:10.3390/jcm11082233_

Round 1

Reviewer 1 Report

Annibali and colleagues performed a well written and comprehensive Review focused on no reflow phenomenon.

The Reviewer thinks that could be useful to provide a step by step practical, but detailed, algorithm to show what kind of medication to use, the suggested dosage and the optimal timing of the use for every drug in this particular condition. 

Author Response

We thank the Reviewer for the comment.

In the revised manuscript, we have added a practical and detailed algorithm for the management of no reflow at our center.

Reviewer 2 Report

The manuscript is well written. The authors summarize the pathogenesis, diagnosis, and treatment of no-reflow. Previous reviews have already demonstrated the no-reflow phenomenon and its pathophysiology, diagnosis, and treatment. With regard to the diagnosis of no-reflow, the diagnostic techniques include angiography, electrocardiography, cardiac MRI, and echocardiography. The author should describe the usefulness of each tool for diagnosis of no-reflow in detail. Moreover, the subjects, the study design, results, and limitations of main references of each technique should also be summarized in table.

Author Response

We thank the Reviewer for the comment.

In the revised manuscript, we expanded the paragraph regarding diagnostic methods and added a summary table at the end. All changes appear underlined in yellow.